# Effects of trust, risk perception, and health behavior on COVID-19 disease burden: Evidence from a multi-state US survey

Benjamin J. Ridenhour[1,2]*, Dilshani Sarathchandra[1,3], Erich Seamon[1], Helen Brown[1,4], Fok-Yan Leung[3], Maureen Johnson-Leon[5], Mohamed Megheib[1], Craig R. Miller[1,6], Jennifer Johnson-Leung[1,2]

1 Institute for Modeling for Collaboration and Innovation, University of Idaho, Moscow, ID, United States of America, 2 Department of Mathematics and Statistical Science, University of Idaho, Moscow, ID, United States of America, 3 Department of Culture, Society and Justice, University of Idaho, Moscow, ID, United States of America, 4 Department of Movement Science, University of Idaho, Moscow, ID, United States of America, 5 Department of Integrative Biology, University of Texas–Austin, Austin, TX, United States of America, 6 Department of Biological Sciences, University of Idaho, Moscow, ID, United States of America

* bridenhour@uidaho.edu

**Data Availability Statement:** The data are available at doi:10.5061/dryad.0cfxpnw4c at datadryad.org.

## Abstract

Early public health strategies to prevent the spread of COVID-19 in the United States relied on non-pharmaceutical interventions (NPIs) as vaccines and therapeutic treatments were not yet available. Implementation of NPIs, primarily social distancing and mask wearing, varied widely between communities within the US due to variable government mandates, as well as differences in attitudes and opinions. To understand the interplay of trust, risk perception, behavioral intention, and disease burden, we developed a survey instrument to study attitudes concerning COVID-19 and pandemic behavioral change in three states: Idaho, Texas, and Vermont. We designed our survey ($n$ = 1034) to detect whether these relationships were significantly different in rural populations. The best fitting structural equation models show that trust indirectly affects protective pandemic behaviors via health and economic risk perception. We explore two different variations of this social cognitive model: the first assumes behavioral intention affects future disease burden while the second assumes that observed disease burden affects behavioral intention. In our models we include several exogenous variables to control for demographic and geographic effects. Notably, political ideology is the only exogenous variable which significantly affects all aspects of the social cognitive model (trust, risk perception, and behavioral intention). While there is a direct negative effect associated with rurality on disease burden, likely due to the protective effect of low population density in the early pandemic waves, we found a marginally significant, positive, indirect effect of rurality on disease burden via decreased trust ($p$ = 0.095). This trust deficit creates additional vulnerabilities to COVID-19 in rural communities which also have reduced healthcare capacity. Increasing trust by methods such as in-group messaging could potentially remove some of the disparities inferred by our models and increase NPI effectiveness.

**Funding:** BJR, JJL received funds via NIH (National Institutes of Health; http://www.nih.gov) Grant number 3P20GM104420-06A1S1. JJL also received intramural funds at the University of Idaho (http://www.uidaho.edu) via the Renfrew Fellowship to pay for undergraduate research. The funders had no role in study design, data collection and analysis, decision to publish, or preparation of the manuscript.

**Competing interests:** The authors have declared that no competing interests exist.

## Introduction

In response to evidence of community spread of COVID-19 [1], the United States (US) began providing guidance and implementing various mitigation policies to reduce disease transmission in March 2020. These mitigation strategies relied on non-pharmaceutical interventions (NPIs) such as mask wearing and social distancing. State and local governments canceled events, issued stay-at-home orders, and mandated the closure of nonessential businesses. Though type, timing, and duration of the orders varied greatly between jurisdictions [2, 3], all of these public health orders called for behavioral changes and restrictions on personal movement, gatherings, and business activity.

In order to properly assess the potential effectiveness of NPIs, decision makers must take into account human behaviors. Voluntary compliance with public health guidance and orders is affected by demographic factors, cognitive constructs, and social constructs [4]. Health behavior theory and risk behavior models [5–7], characterize the demographic factors related to risk perception and health protective behavior. The perception that viruses pose a serious threat, and that one is susceptible to this threat, is the most likely predictor of adoption and compliance with NPIs. Other cognitive constructs, namely perceived severity, perceived susceptibility, and belief in the benefits of adopted behaviors, are all associated with reduced COVID-19 risk behaviors and increased health protective behavior [8, 9]. That behaviors are often shaped by perceptions of what others are doing, by in-group approval, and by desires to protect those in their communities [10, 11] further complicates NPI adoption.

Political identity in particular is one factor that can lead to out-group distrust [12]. Affective polarization [13] extends beyond issue-based disagreement to an identity-based comparison between in-groups and out-groups. This exacerbates dislike and distrust of those outside of the in-group [13–15]. When public health institutions are considered part of the out-group, this can result in non-adoption of preventative behaviors [16].

Rural Americans face increased risk of severe illness and death from COVID-19 due to health disparities, health care shortages, and social inequities [17, 18]. On average, rural Americans are older, are more likely to live in poverty, have higher rates of chronic disease and disability, and are less likely to be insured than urban dwellers [19, 20]. Studies have consistently shown less compliance with NPIs in rural areas, particularly among rural Americans identifying as conservative. These associations were less strong among older rural individuals [17, 18]. The lack of healthcare resources due to hospital closures, limited numbers of health professionals, and low critical-care capacity in rural communities poses an additional risk in the face of a surge of patients with COVID-19 [21].

In this study, we use a survey instrument distributed in three socially and demographically diverse US states (Idaho, Texas, and Vermont) during October and November 2020 in order to examine the differences among rural and urban Americans in their attitudes towards, and uptake of, NPIs. To advance health behavior theory, we tested various causal relationships between trust in public health guidance, health and economic risk perception, and resistance to pandemic behavioral change using structural equation modeling. Secondarily, we also explore the relationship between disease burden and behavior with models of our survey data. From the best-supported models, we determine how rurality—along with other exogenous variables such as political ideology—factors into behavior during the early portion of the COVID-19 pandemic. We emphasize that our model is not an attempt to produce the best predictive model of COVID-19 burden, an effort which has been done using many other better suited methods for that task. Rather, we wish to determine a model of human behavior that could augment such models and increase their value to public health officials. Our work is important and novel because it incorporates human attitudes, perceptions, and behavioral

intention into infectious disease models, which extends our ability to predict expected differences in disease outcomes across the United States.

## Materials and methods

### Survey development and data collection

Data for this research come from a sequential mixed-mode survey distributed to a disproportionate stratified sample of households in Idaho, Texas, and Vermont. The specific survey design, employing both an online and a paper survey option, as well as English and Spanish translations, was selected in order to reach communities that are typically harder to reach via online surveys (for example, rural and elderly populations, individuals who lack access to reliable internet connections, and non-English speakers) [22]. Following standard survey design principles, the survey design includes several steps: pre-testing, field testing, pilot testing, and validation [23, 24]. We first pre-tested the survey with a convenience sample of college students ($n = 55$) recruited from the University of Idaho via the online survey platform Qualtrics (Provo, UT, USA). Pre-testing enabled us to measure pertinent factors such as time for completion, satisfaction, and level of difficulty. Subsequently, we field tested the survey questionnaire by sharing it with 10 state and regional public health experts and one community based organization serving Hispanic populations in Idaho (Community Council of Idaho). This organization helped us to verify the Spanish translation and determine its cultural resonance. Feedback from these experts was used to revise and refine the survey questions to help ensure their validity and reliability. Lastly, we pilot tested the survey using Qualtrics by distributing the survey to 50 respondents each from ID, TX, and VT ($n = 150$) between August-September 2020. For each state, we obtained equal proportions of rural and urban/suburban respondents. We conducted consistency analysis using the pilot data and examined other factors such as time for completion and any inexplicable patterns in the pilot data. The finalized survey covers topics including: worry about COVID-19, social distancing, mask wearing, economic impacts, contact tracing, vaccination intention, trust, information sources, and demography. Our questions are theory driven, tapping into constructs from common health behavior theories such as Social Cognitive Theory [25] and the Health Belief Model [26]. We also rely on CDC's Behavioral Risk Factor Surveillance System (BRFSS) and other published survey studies, e.g., Jamieson and Albarracín [27], to determine the consistency and validity of survey questions.

Our disproportionate stratified sample purchased from Dynata (Shelton, CT, USA) consists of 2000 rural and 2000 urban or suburban addresses from each of Idaho, Texas, and Vermont (12000 in total). Dynata classifies addresses as rural if they fall outside of a metropolitan statistical area (MSA) as defined by the US Office of Management and Budget. We employed the services of Washington State University's Social & Economic Sciences Research Center to distribute the survey. All household addresses within the sampling frame were sent an initial invitation letter—which included a $1 USD incentive—directing respondents to a URL where they were asked to enter their unique response ID and complete the survey online. Non-respondents were sent a reminder postcard one week later, and two weeks after that a final reminder letter was mailed. We offered a phone number and an email address with the option to reach out to us to request a paper copy of the survey for those preferring the paper option. Online survey data collection occurred during October and November, 2020. Requested paper surveys were mailed in mid-November, and the data collection was completed in December, 2020.

Our survey questionnaire (S1 Appendix) was approved by the University of Idaho Institutional Review Board (IRB #20–119). This study was deemed exempt from full review by the IRB as it includes a voluntary survey data collection of adults over the age of 18. Informed consent was obtained from all survey participants. Consent was documented by online survey

participants reading the consent form and voluntarily clicking a button to proceed to the full survey. The participants who took the paper survey read a consent form and voluntarily mailed back their completed surveys. This study does not include any retrospective medical records or archived samples.

## Measurements

Our demographic variables are comprised of direct measures of five attributes. Political ideology is coded as an unordered factor with levels: liberal, moderate, conservative, libertarian, non-political, and other; moderate is designated as the reference level for statistical analyses. The remaining measures are recorded as Boolean variables measuring race (white = 1), gender (female = 1), age (over 64 years = 1), and geography (rural = 1). See S3.1 Table in S3 Appendix for a detailed breakdown of demographic characteristics. Except for geography, which is determined by our de-identified address-based survey sample, all demographic variables are self-reported.

Rural/urban designations for each response are determined based on the United States Department of Agriculture's (USDA) rural-urban commuting area codes (RUCA), which classify US census tracts based on population density, urbanization, and daily commuting distance [28]. While RUCAs utilize a similar metropolitan/micropolitan approach used as part of the Office of Management and Budget (OMB) classification of metropolitan statistical areas (MSAs), the use of census tracts in RUCA assignment provides a more detailed geographic structure for urban and rural delineation [29].

For our analyses, we geographically mapped all survey respondents ($n$ = 1034) for all three states (ID, TX, and VT), and associated RUCA codes based on the respondents' de-identified addresses. We used ArcGIS software from Environmental Systems Research Institute, Inc. (ESRI; West Redlands, CA, USA) to perform this spatial association. We then designated respondents whose RUCA primary code was 1, 2, 3, or 4 as urban, and all other RUCA codes as rural (see S3.2 Table in S3 Appendix for a full list of all RUCA code designations). This stricter classification, as opposed to MSA classifications used in the sampling frame, ensures that rural-classified responses would reflect rural attitudes and experience [29].

We use two different measures of disease burden. For models where behavioral intention is hypothesized to affect disease burden, we consider cumulative cases per 100 people from the beginning of the pandemic in January 2020 through 30 April 2021, at the county level, as reported by the New York Times [30]; county-level data represent the finest spatial scale available for use in the study region (e.g., city-level data are not available). Choosing a date after the survey period enables observation of delayed consequences of behavior on disease burden. The chosen sample date captures the main wave of the pandemic in the US prior to widespread availability of the vaccines. Exploratory analyses showed that the exact choice of date has little-to-no effect on model results, which is to be expected given the auto-correlative structure of spatiotemporally-distributed cumulative disease data. For models where disease burden is hypothesized to affect behavioral intention, we use the cumulative case count from January 2020 to the recorded response date of the observation. Using this measure is consistent with the idea that previous, personal experience with the pandemic is shaping behavior. For all models, each respondent is assigned disease data corresponding to the county of the sampled de-identified address.

We explored using other measures of burden, in particular the number of COVID-19 deaths reported. Death data are highly correlated with case data, which produced model convergence issues (singularities) if both measures were used simultaneously. Use of death data alone produced only slight changes in our model results, and thus are not presented herein.

Because much of sample comes from small rural populations, death data are sparse due to increased stochasticity; this sparsity reduces statistical power, and we therefore opted to use cumulative case counts.

## Statistical models

We use structural equation modeling (SEM) to test six different hypotheses regarding potentially causal pathways between trust, two types of risk perception, behavioral intention, and disease burden. SEM is a flexible modeling method that allows for the decoupling of the random error arising from observation and the error in the model and comparison of different pathways of causality. Our SEM application utilizes the lavaan package [31] which integrates factor analysis to define the latent variables with systems of simultaneous linear regression equations. Latent variables are those for which there is no direct measurement, but rather the variables are inferred indirectly via indicators. All analyses were performed in R v4.1.0 (see S2 Appendix for R code). The four central latent variables in our attitudinal framework are inferred as follows.

"Health risk perception" and "economic risk perception" are derived from survey questions which asked directly about respondents' concern for their own and community health and economic security. For these two measures, higher values of the latent variable indicate higher perceived risk. "Trust" is derived from similar questions which probe the degree of trust in COVID-19 guidance from governmental public health, medical, and scientific authorities. For our trust measure, higher values indicate higher trust in selected sources.

"Behavioral intention" is derived from 8 other latent variables corresponding to expected engagement in day-to-day activities and protective behavior. Specifically, these activities include: 1) gathering indoors with close friends and family, 2) gathering indoors with a large group, 3) dining indoors at a restaurant, 4) attending church indoors, 5) shopping in person, 6) attending personal appointments, 7) participating in large community activities, and 8) wearing a mask. Questions about participating in these 8 activities were presented at increasing COVID-19 exposure-risk levels. A higher behavioral intention score indicates that the respondent expects to continue activities 1–7 and eschew masking as risk levels increase.

Health risk perception, economic risk perception, and trust, together with the second-order variable behavioral intention, form our attitudinal framework. See S3.2 and S3.3 Fig in S3 Appendix for specifics of the latent variable submodels. For all of the latent variables in our attitudinal framework, as well as disease burden, we controlled for demographic variables via structural regressions (rural/urban, female/non-female, white/non-white, over/under 65 years old, and political ideology). Other control variables—such as education and income—were originally explored in structural regressions as well; however, due to a lack of significant impact, these variables have been dropped from the presented analyses.

Prior studies have considered various causal relationships between trust and risk perception [32], finding support for influence in both directions depending on the context. In order to determine the best fitting causal framework for this study, we test three different relationships between them: 1) trust affecting risk perception (models 1*A* and 1*B*), 2) risk perception affecting trust (models 2*A* and 2*B*), and 3) independence of trust and risk perception (models 3*A* and 3*B*). We also test the direction of the causal relationship between disease burden and behavioral intention (e.g., by comparing model 1*A* to 1*B*). Thus, we test six different competing hypotheses using SEM in total. Fig 1 gives graphical representations of the different hypotheses being compared. For all of the factor analyses and structural regressions being run, the observational unit is at the individual-level with the exception of the 3 structural regression models (one each in models 1A, 2A, and 3A) where the dependent variable is cumulative cases

**MODEL 1: TRUST TO RISK TO OUTCOME**

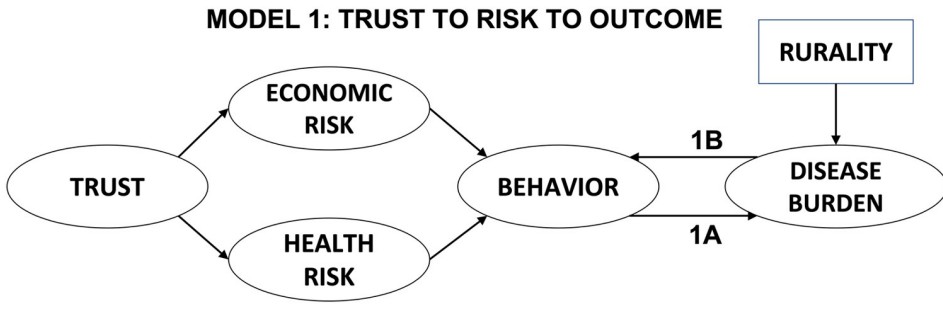

**MODEL 2: RISK TO TRUST TO OUTCOME**

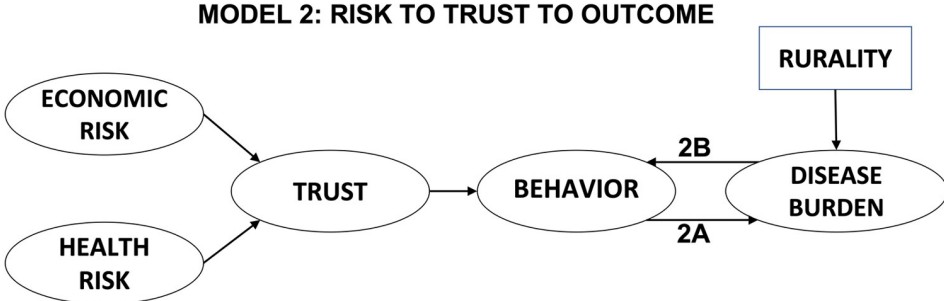

**MODEL 3: TRUST AND RISK TO OUTCOME**

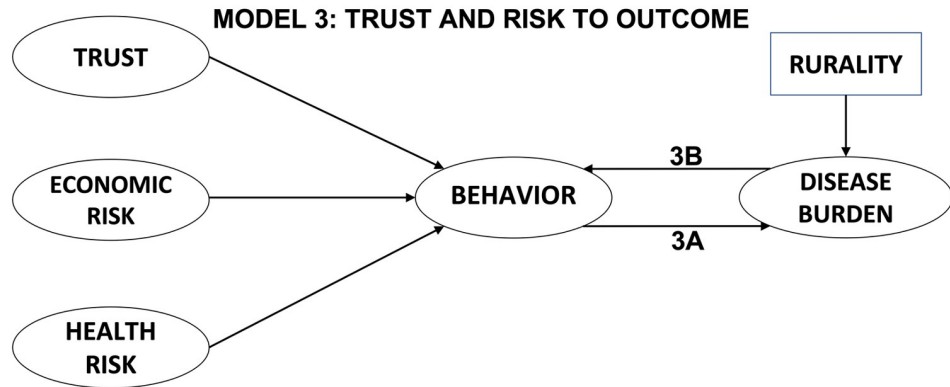

**Fig 1. Hypothesized conceptual models.** We tested several hypotheses about the interplay between trust, risk perception, behavioral intention, and COVID-19 disease burden. Each path diagram shows the hypothesized causal relationships between our measured variables. Latent variables are shown in ovals; exogenous variables are shown in rectangles. Structural equation modeling (SEM) was used to assess which model was best supported by our survey data.

per 100; in these three models, 117 unique values of case burden as of 30 April 2021 were available for the 117 counties sampled and were paired according to each individual's county of residence.

## Results

Overall, we received 1087 responses, a majority online. 57 people chose to receive a paper copy of the survey, and 44 of those respondents mailed completed surveys back to us. Our overall response rate is 9.98%, excluding the 1110 addresses that were not deliverable. After

eliminating redundant and incomplete surveys, we received 1034 responses that were usable for analysis. Raw data are available at Dryad (http://datadryad.org); S2 Appendix contains R scripts with our data processing and analysis routines.

Of the survey respondents, 55% identify themselves as female, 44% as male, and 0.6% as neither male nor female. The mean age of the full sample is 55 years of age (range 16–96, SD = 16.42). A majority of our sample has college degrees or higher levels of education (66%) followed by those who have attended some college (16%). A majority indicate that their total household income exceeds $75,000 per year (52%) with only 7% reporting household incomes less than $25,000. In terms of ethnicity, 4% of our respondents are Hispanic or Latino; racially, the majority of respondents were white (85%). Approximately 29% of respondents in each category identify as liberal, moderate, and conservative while the rest identify as Libertarian, non-political or other. Most of our respondents indicate that they are currently married or in domestic partnerships (68%). In terms of religion, most respondents identify as evangelical Christian (17%), followed by Catholic (16%), Mainline Christian (15%) and Agnostic (14%). S3.1 Table in S3 Appendix has a full breakdown of our demographic variables. Overall, our survey sample is disproportionately white, has higher levels of education and income, and is older than the national and state distributions, which has been observed elsewhere in mail surveys [33].

We tested whether demographic distribution of our responses is dependent on the state in which a respondent lives. Overall, respondent distributions for age, gender, and income are similar across ID, TX, and VT. While our sample has a large fraction of rural responses due to the sampling method, the only state for which a majority of respondents are rural is Vermont; the distribution of urban/rural respondents is significantly different between the sampled states ($\chi^2$: 46.74, df: 2, $p < 0.001$). Statistically significant differences are also observed for political orientation ($\chi^2$: 113.04, df: 10, $p < 0.001$), ethnicity ($\chi^2$: 29.45, df: 2, $p < 0.001$), race ($\chi^2$: 71.64, df: 14, $p < 0.001$), educational attainment ($\chi^2$: 31.37, df: 8, $p < 0.001$), relationship status ($\chi^2$: 24.3, df: 8, $p = 0.002$), and religion ($\chi^2$: 201.83, df: 18, $p < 0.001$).

For our SEM models, of the 1034 respondents, 829 are usable ("complete" or lacking any missing columns) for this analysis. All of the SEM hypotheses in which we test behavioral intention driving disease burden produce good fits (RMSEA values of 0.071, 0.072, and 0.072, respectively for models 1A, 2A, and 3A). Comparison of models 1A, 2A, and 3A via Akaike's Information Criterion (AIC) give values of 98774, 98950, and 98932, respectively. Likelihood ratio tests indicate that the first model, where trust affects risk perception, is supported significantly better by our data (model 1A vs. 2A: $\chi^2 = 178.41$, df = 1, $p < 0.001$; model 1A vs. 3A: $\chi^2 = 159.91$, df = 1, $p < 0.001$). Thus, all fit measures indicate 1A to be the best supported model of the three.

For models 1B, 2B, and 3B, where behavioral intention is hypothesized to be affected by prior pandemic experience, we also observe good fits of the model (RMSEA values of 0.071, 0.072, 0.072, respectively). AIC values for these models are 89908, 90122, and 90080, respectively, and likelihood tests again favor the first trust-risk structure (model 1B vs. 2B: $\chi^2 = 216.12$, df = 1, $p < 0.001$; model 1B vs. 3B: $\chi^2 = 173.51$, df = 1, $p < 0.001$). Thus, model 1B is the best supported model of the three by all fit measures. From hereon, we will focus on the results of models 1A and 1B for the remainder of this article. We note that there is no method to *statistically* compare model 1A with 1B because of differences in the underlying data and equation structures. However, because the model structures for the social and cognitive latent variables in these models are, in general, similar, we report the p-values in parallel with $p_A$ denoting the p-value of model 1A, and $p_B$ denoting the p-value of model 1B; the full results of each of the models are provided in S3.3-S3.8 Table in S3 Appendix.

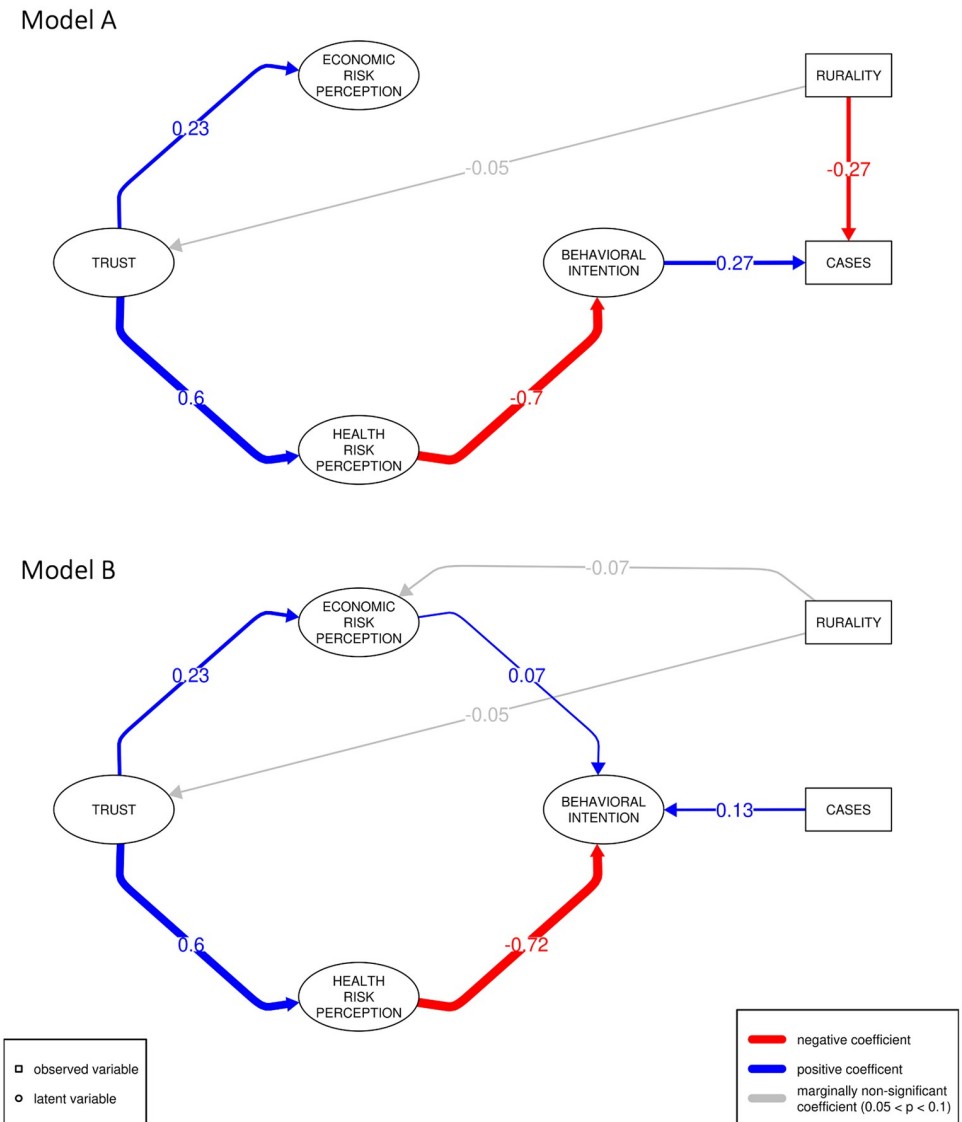

**Fig 2. Best supported social cognitive model with *A* and *B* types.** The results of our SEM show that the model where trust influences perceived health risk which in turn alters behavioral intention is the best of our causal hypotheses. Model *A* shows behavioral intentions affecting disease burden, and Model *B* shows the effect of disease burden on behavioral intention. In model *A*, rurality has an indirect effect on disease burden, with a negative effect on trust ultimately leading to increased disease burden. Only pathways that are supported with *p* < 0.1 are shown.

Both model 1*A* and model 1*B* posit that trust affects risk perception which subsequently affects behavioral intention; for model 1*A* all paths are significant with the exception of economic risk perception affecting behavioral intention, while for model 1*B* all paths are significant (Fig 2). Increased trust leads to increased health risk perception ($p_A$, $p_B$ < 0.001) and economic risk perception ($p_A$, $p_B$ < 0.001). Higher health risk perception is associated with lower behavioral intention to engage in activities that have greater potential for exposure to disease ($p_A$, $p_B$ = 0.009). For model 1*A*, this riskier behavioral intention leads to increased disease burden ($p_A$ = 0.013). While for model 1*B*, where we consider the impact of the respondents' pandemic experience on their expressed behavioral intentions, higher disease burden is

associated with riskier behavioral intention ($p_B = 0.019$). In model 1B, we also found a marginally significant effect of increased economic risk perception on behavioral intention ($p_B = 0.073$).

In model 1A, we can consider the indirect impacts of social cognitive factors on disease burden via behavioral intentions. For the indirect effect of trust on disease burden, mediated via perceived health risk and behavioral intention, we find that increased trust is significantly associated with decreased disease burden ($p_A < 0.001$); the intermediate pathway of increased health risk perception is also associated with higher disease burden ($p_A < 0.001$). Because rurality has a marginal effect on trust, we examine the indirect effect of rurality on disease burden mediated via the trust-health risk-behavioral intention pathway. We find that this indirect effect of rurality increases disease burden but is only marginally significant ($p_A = 0.095$). However, the net effect of rurality is still protective at the time of the survey because rural areas experienced fewer COVID-19 cases per capita through the Spring of 2021, i.e., the direct effect of being rural overwhelmed the indirect effect.

The results of the factor analysis for behavioral intention indicate which activities are tied to a higher intention to engage in activities that potentially increase exposure to COVID-19. Recall that behavioral intention is estimated using 8 day-to-day activities as indicator variables. The results of the SEM for both models show that all 8 activities are at least marginally significant for this measure. (Only respondents' answers regarding willingness to go shopping and attend appointments are marginally significant.) Listed from strongest association to weakest association, indicators of behavioral intention are eating in restaurants, attending indoor group gatherings, participating in large community activities, attending church, going to appointments, meeting indoors with close friends and family, mask wearing, and shopping.

We find that demographic variables have significant effects on several of the latent variables. The only significant effect of geography is on disease burden, with rural communities having a significantly lower disease burden ($p_A < 0.001$). There are, however, marginally significant effects of rurality on trust (decreasing; $p_A$, $p_B = 0.088$) and on economic risk perception (decreasing; $p_A = 0.096$, $p_B = 0.094$). Women show significantly increased health risk perception ($p_A$, $p_B < 0.001$) and economic risk perception ($p_A$, $p_B = 0.035$). Women are also significantly more likely to continue daily activities in 1A ($p_A = 0.045$); this effect was marginally significant in 1B ($p_B = 0.057$). Individuals who are white have significantly higher trust ($p_A$, $p_B < 0.001$) and lower perceived health risk ($p_A$, $p_B < 0.001$). Elderly individuals perceive significantly higher health risk ($p_A$, $p_B < 0.001$).

The most significant exogenous factor included from our survey data is political ideology. Compared to respondents self-identifying as moderates, self-identified liberals communicate more trust ($p_A$, $p_B = 0.001$) and self-identified conservatives communicate the least trust ($p_A$, $p_B < 0.001$). Those self-identifying as non-political or libertarian also express significantly less trust than self-identified moderates ($p_A$, $p_B < 0.001$ for both). In terms of risk to health from COVID-19, self-identified liberals are significantly more concerned ($p_A$, $p_B = 0.013$), while self-identified libertarians are less concerned ($p_A = 0.053$, $p_B = 0.050$), though this effect is marginal in model 1A. In considering economic risks from the pandemic to themselves and their community, self-identified conservatives are less concerned than self-identified moderates ($p_A$, $p_B = 0.005$), while self-identified libertarians are more concerned ($p_A$, $p_B = 0.023$). Finally, with respect to behavioral intention, identifying as conservative has the strongest positive association with increased behavioral intention to continue pre-pandemic activities and avoid masking ($p_A = 0.013$, $p_B = 0.014$); self-identified libertarians are also more likely to take on more risk of exposure to COVID-19, though only marginally so ($p_A = 0.079$, $p_B = 0.084$). In model 1A, self-identified liberals are predicted to have increased protective behavior ($p = 0.037$), but not in model 1B.

## Discussion

Our results imply there are downstream, indirect consequences of demographic and ideological characteristics on behavior and potentially on disease burden. Specifically, we find the most support for social cognitive models where trust influences risk perception, which in turn affects behavioral intention (models 1*A*, 1*B*). Counter intuitively, model 1*B* predicts higher observed disease burden during the beginning of the COVID-19 pandemic is associated with decreased prophylactic behaviors. Individuals from rural communities express reduced trust and reduced perceived risk, indicating that the barrier to public health engagement is stronger in these regions. Importantly, our research suggests cultivating trust in authorities tasked with communicating public health information would be the optimal way to increase adoption of NPIs to slow the spread of future pandemics.

In the case of COVID-19, trust in the message and the messenger has been undermined by several factors. Namely, there was a lack of uniform national, state and local strategies; inadequate reach, accessibility, and consistency of public health information; and widespread misinformation and disinformation that was not adequately refuted [34, 35]. Studies suggest that misinformation not only erodes trust in public health authorities, but also decreases the motivation to seek and adopt correct information [36]. The influence of social media on information consumption exacerbates the impact of misinformation [27]. News partisanship further impacts trust in public health authorities' message of risk and the reduction of risk through social distancing and other actions [37, 38].

COVID-19 pandemic response protocols ask individuals, families, schools, and communities to adopt life-altering precautions and behavioral changes. To adopt these practices individuals must perceive the risk of COVID-19 to themselves, their families, and communities. Furthermore, they must trust public health authorities to accurately identify and communicate protective disease intervention protocols [39]. One consequence of the request by authorities for social distancing and mask wearing was increased uncertainty and skepticism [40, 41]. Individuals with more trust in public health authorities are less likely to characterize such requests as a result of incompetence or malfeasance and comply [35]. The result of increased trust leading to increased pandemic protective behavior, as measured by decreasing day-to-day activities and increasing mask wearing, is borne out in both of our best supported models (Fig 2).

Observed early support for NPIs in the US was notably absent in rural communities and essential workers [42]. Our analyses similarly shows lower levels of institutional trust, lower levels of intention to comply with public health measures, and decreased risk perception in rural areas (Fig 2). Nonetheless, our analyses also shows that disease burden was significantly lower among rural persons. This suggests that, at least in the earlier stages of the pandemic, rurality had a protective effect. This was most likely due to reduced population density and time-of-onset of epidemic waves in those areas. However, not all rural residents were at low exposure risk to SARS-CoV-2. Some rural residents working in meat, poultry, food processing, and agricultural industries face additional COVID-19 risks as these industries involve working and/or traveling in enclosed spaces closer than the recommended 6-foot distance. These industries were deemed essential and were not closed, even in cases of high community transmission. As a result, outbreaks of COVID-19 disproportionately impacted workers and their families in such industries [43]. Decreased levels of trust in rural areas likely worsened the issues stemming from these outbreaks among essential workers.

Our best supported models propose a role for behavioral intention in influencing future disease burden (model 1*A*) and, conversely, previously observed disease burden influencing behavioral intention (model 1*B*). In model 1*A*, we find that resistance to behavioral change

during the COVID-19 pandemic, vís-a-vís adoption of NPIs, is significantly predictive of higher disease burden in the Winter 2020 wave of the COVID-19 pandemic. This result fits with standard epidemiological theory where the rate and frequency of uptake of NPIs drastically affects the epidemic trajectory. Classic examples of these effects are found in post-hoc analyses of the 1918 Spanish Flu pandemic [44, 45]. Surprisingly in model 1*B*, we find that increased observed prior disease burden actually leads to reduced prophylactic behavior. While this result is counter intuitive, it is perhaps not without precedent. Recent research [46, 47] shows that *perceived* disease severity is influenced by various ideological and social factors. Therefore, one potential explanation for the predictions of model 1*B* would be a disconnect between perceived and actual disease burden in a county. If individuals are being told by their in-group that disease burden is not severe, then they may continue engaging in behaviors that increase their chances of contracting COVID-19, even in the face of high case counts. These effects may have been worsened by the fact that a majority of COVID-19 cases are mild and deaths are concentrated in the elderly [48, 49].

In the United States, adoption and approvals of public health interventions for COVID-19 fall along political lines. Specifically, other research finds people identifying as democrats favor publicly mandated disease interventions and practice protective health recommendations more than people identifying as republicans [37, 50–52]. Political ideology similarly influences every aspect (trust, risk perception, and behavioral intention) of our social cognitive model results. The finding that political ideology affects trust and compliance with NPIs (i.e., behavioral intention in our models) has been reported in several other studies [34, 37, 41, 53]. Furthermore, our findings are consistent with other recent work in which partisan differences were found to be more significant than other factors in determining social distancing behavior, and with results of disparate health outcomes based on party identity [38, 54]. Thus, our work adds to the body of evidence for the consequences of political ideology on behavioral changes in response to the pandemic.

Our model, however, offers a more nuanced view of where partisanship plays a role in affecting various aspects of cognition. In particular, the social construct of trust in public health guidance seems to be affected by all of the political categories we analyzed (i.e., liberal, moderate, conservative, libertarian, non-political). For the cognitive constructs, only libertarian identity is significant for both health and economic risk perception. In addition, health risk perception is also significantly affected by liberal identity, while economic risk perception is significantly affected by conservative identity. Lastly, self-identified liberals expressed willingness to reduce their day-to-day activities as the risk of SARS-CoV-2 infection increased, while conservative and libertarian identities were significantly associated with reluctance to reduce activity. Therefore, public health strategies appealing to certain cognitive constructs might be better focused toward particular partisan groups. For example, advertising health risks of a disease may impact liberals and libertarians more effectively than other groups. Still, trust has the strongest effect on both types of risk perception, there we suggest maintaining trusting relationships with all groups is the most vital action.

Our findings related to gender are also in-line with other studies that report women as more concerned about the health consequences of COVID-19 [55–57]. These results are somewhat surprising given that men are more likely to contract severe COVID-19 cases resulting in hospitalization or death [58]. However, our findings that women engage in higher levels of activity that could expose them to SARS-CoV-2 differ from other studies [56]. This might be explained partially by the increased household responsibilities of women resulting in higher activity levels [59]. 64% of the women in our survey indicated that they are married and therefore may feel increased pressure to perform some the day-to-day activities about which we asked. Finally, women also perceived more economic risk to themselves and their community

from COVID-19, which is consistent with women having generally higher risk perception [60].

Our study has several limitations. First, survey instruments are subject to response bias. Our respondents tend to be older, wealthier, more-educated individuals compared to the population as a whole. This is typical of many survey-based studies [33]. We interpret our findings in light of this limitation. Second, we received fewer responses from Texas (144) than from Idaho and Vermont. However, we received a substantial fraction of rural responses from each state, resulting in a multifaceted picture of rural attitudes; therefore, the effect of a lack of respondents from Texas may have been minimal. Third, with respect to disease data, we are limited by the shortcomings of the disease surveillance and reporting mechanisms. Because of limitations in testing for COVID-19, reported case counts are an underestimate of the true number of cases. This should have little effect on the outcome of our study so long as there are no heterogeneous biases in under-reporting of cases. Fourth, we are limited by the fact that COVID-19 cases are reported at the county level within the US. We may have been able to achieve greater resolution in our study had we been able to associate case counts with census tracts, the geographic level at which the geographic analysis was conducted. Related to this, in determining whether a zip code is rural or urban, we use the RUCA classification system. This system offers a finer level of granularity of which locations are urban and which are rural than the MSA classifications used by Dynata. Fifth, our survey instrument measures an individual's self-reported political leanings, rather than political affiliation directly. Previous work shows that individuals may be afraid to honestly identify their political beliefs for fear of repercussions. Sixth, it should be emphasized that our study represents a snapshot of attitudes in late 2020, and it is possible that attitudes toward NPIs have changed with the progression of the pandemic and the availability of effective vaccines. Finally, while we received 1,087 responses out of the 12,000 surveys we sent out, having a larger sample size may have allowed us to attribute significant effects to other factors than those discussed here. That being said, the smallest significant effects in our models are of magnitude around 0.05, which suggests that our analyses are strong enough to detect small effects.

## Conclusion

Understanding how individuals process and respond to threats in their environment is critical to optimizing public health messaging and policy. Using structural equation modeling to identify latent variables for trust, risk perception, and behavioral intention, our survey results best support the hypothesis that building trust in government organizations can be used to influence behavioral intentions indirectly via risk perception. Higher risk perception leads to reduced behavioral intention, and model 1*A* predicts reduced behavioral intentions leads to reduced disease burden. We therefore propose decision makers focus efforts on trust building to increase NPI effectiveness in future pandemics. Our work is novel in its attempt to reach and understand individuals living in rural areas. Rural populations indicate less trust and reduced risk perception compared to urban populations, making them vulnerable to higher disease burden and a possible focus area for public health. Lack of trust in rural communities combined with increased risked to essential workers could have negative synergy; this issue is beyond the scope of this work but merits future study. In agreement with other COVID-19 studies, political ideology seems to be an overwhelming factor influencing the trust–risk–behavior cognitive pathway. Our results align with other research on politicization and polarization of public views towards controversial topics. Future research utilizing increased spatial and temporal resolution of survey data, along with other measures of disease burden, such as years-of-life-lost, could further elucidate the links between political affiliation and social cognition.

## Supporting information

**S1 Appendix. Full English survey.**
(DOCX)

**S2 Appendix. R markdown file with detailed R code.**
(RMD)

**S3 Appendix. Compiled version of R markdown file.**
(PDF)

## Acknowledgments

We would like to thank the pandemic modeling group at the Institute for Modeling Collaboration and Innovation (IMCI) at the University of Idaho for help working on and thinking about COVID-19 related issues. Similarly, we thank the University of Texas–Austin COVID modeling consortium, led by Drs. Lauren Ancel Meyers and Spencer Fox, for useful conversation and feedback regarding this work. Specific thanks to individuals at the University of Idaho go out to our undergraduate research assistants Isabella Bermingham, Chloe Dame, Maria Elizarraras, and Bishal Thapa, and Drs. Erkan Buzbas and Tim Johnson for helpful conversations about statistical models. Finally, we thank Dr. Holly Wichman for her invaluable research leadership at IMCI.

## Author Contributions

**Conceptualization:** Benjamin J. Ridenhour, Dilshani Sarathchandra, Helen Brown, Maureen Johnson-Leon, Mohamed Megheib, Craig R. Miller, Jennifer Johnson-Leung.

**Data curation:** Erich Seamon, Mohamed Megheib, Jennifer Johnson-Leung.

**Formal analysis:** Benjamin J. Ridenhour, Dilshani Sarathchandra, Erich Seamon, Fok-Yan Leung, Mohamed Megheib, Craig R. Miller, Jennifer Johnson-Leung.

**Funding acquisition:** Benjamin J. Ridenhour.

**Methodology:** Benjamin J. Ridenhour, Dilshani Sarathchandra, Erich Seamon, Helen Brown, Fok-Yan Leung, Maureen Johnson-Leon, Mohamed Megheib, Craig R. Miller, Jennifer Johnson-Leung.

**Project administration:** Jennifer Johnson-Leung.

**Resources:** Fok-Yan Leung, Jennifer Johnson-Leung.

**Software:** Erich Seamon.

**Supervision:** Benjamin J. Ridenhour.

**Writing – original draft:** Benjamin J. Ridenhour, Dilshani Sarathchandra, Erich Seamon, Helen Brown, Fok-Yan Leung, Maureen Johnson-Leon, Mohamed Megheib, Craig R. Miller, Jennifer Johnson-Leung.

**Writing – review & editing:** Benjamin J. Ridenhour, Dilshani Sarathchandra, Erich Seamon, Helen Brown, Fok-Yan Leung, Maureen Johnson-Leon, Jennifer Johnson-Leung.

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
