## [Decision Letter · Decision Letter 0]

28 Dec 2021

PONE-D-21-36310Effects of trust, risk perception, and health behavior on COVID-19 disease burden: Evidence from a multi-state US surveyPLOS ONE

Dear Dr. Ridenhour,

Thank you for submitting your manuscript to PLOS ONE. After careful consideration, we feel that it has merit but does not fully meet PLOS ONE’s publication criteria as it currently stands. Therefore, we invite you to submit a revised version of the manuscript that addresses the points raised during the review process.

We look forward to receiving your revised manuscript.

Kind regards,

José Alberto Molina

Academic Editor

PLOS ONE

Journal Requirements:

(We would like to thank the pandemic modeling group at the Institute for Modeling Collaboration and Innovation (IMCI) at the University of Idaho for help working on and  thinking about COVID-19 related issues. Similarly, we thank the University of  Texas–Austin COVID modeling consortium, led by Drs. Lauren Ancel Meyers and Spencer Fox, for useful conversation and feedback regarding this work. We also thank our undergraduate research assistants Isabella Bermingham, Chloe Dame, Maria Elizarraras, and Bishal Thapa who were supported by a College of Science Renfrew  Faculty Fellowship awarded to JJL. We thank Drs. Erkan Buzbas and Tim Johnson for  helpful conversations about statistical models. Finally, Dr. Holly Wichman has  provided invaluable leadership in finding and giving funding to perform this research;  specifically, this work was funded by NIH Grant number 3P20GM104420-06A1S1.)

(BJR, JJL received funds via NIH (National Institutes of Health; http://www.nih.gov) Grant number 3P20GM104420-06A1S1. JJL also received intramural funds at the University of Idaho (http://www.uidaho.edu) via the Renfrew Fellowship to pay for undergraduate research.  The funders had no role in study design, data collection and analysis, decision to publish, or preparation of the manuscript.)

4. In your Data Availability statement, you have not specified where the minimal data set underlying the results described in your manuscript can be found (S2 R markdown file with detailed R code). PLOS defines a study's minimal data set as the underlying data used to reach the conclusions drawn in the manuscript and any additional data required to replicate the reported study findings in their entirety. All PLOS journals require that the minimal data set be made fully available. For more information about our data policy, please see http://journals.plos.org/plosone/s/data-availability.

Additional Editor Comments:

The literature about the Covid effects is very extensive, e.g https://covid-19.iza.org/publications/ and, similarly, with other wp series and, of course, journals. It is particularly relevant to review the economic papers which use this kind of econometric methods to evaluate the covid-19 effects. Authors need to prove the novelty of this contribution, given that ethods are not particularly solid. Consequently, It is absolutely needed to highlight the novelty, in addition to perform the methods in a solid way.

Reviewers' comments:

Reviewer's Responses to Questions

**Comments to the Author**

1. Is the manuscript technically sound, and do the data support the conclusions?

Reviewer #1: Partly

Reviewer #2: Partly

Reviewer #3: Yes

2. Has the statistical analysis been performed appropriately and rigorously? 

Reviewer #1: Yes

Reviewer #2: Yes

Reviewer #3: Yes

3. Have the authors made all data underlying the findings in their manuscript fully available?

Reviewer #1: Yes

Reviewer #2: Yes

Reviewer #3: Yes

4. Is the manuscript presented in an intelligible fashion and written in standard English?

Reviewer #1: Yes

Reviewer #2: Yes

Reviewer #3: Yes

5. Review Comments to the Author

Reviewer #1: Referee report for “Effects of Trust, Risk Perception, and Health Behaviour on Covid-19 Disease Burden: Evidence from a Multi-State US Survey

Summary:

This paper collects survey evidence from around 1000 individuals in Texas, Idaho and Vermont in November 2020, measuring demographic characteristics, social behaviours, risk perceptions, trust and political ideology. The paper assesses the relationship between these variables and case counts using SEM modelling. The paper finds that lower trust in rural locations offsets the natural advantages these places have for combating the spread of Covid-19.

Assessment

My main technical concern is with the implementation of the model. In many cases (e.g. Model 1A) the authors model behavioural intention as determining cases. The behavioural intention is then measured at the individual level, while case numbers are measured in the aggregate. But of course an individual’s behaviour is atomistic and has negligible effect on aggregate case numbers. So I don’t know how to interpret the estimates. I can’t recall ever seeing a model with an aggregate explained variable being driven by an individual-level explanatory variable. This approach should at least be discussed and supported.

More subjective concerns are as follows:

The main problem with this paper is the extremely small sample size. In each model the authors use 1000 observations to estimate 40 parameters. Obtaining significant and novel results in this way is only possible with rigid modelling.

My technical concern above leads me to another criticism, that much of questions here would be better addressed using aggregate data variation over both location and time. For example the google data can be leveraged to obtain detailed aggregate measures of behaviour across locations and over time. Similarly social surveys could be leveraged to obtain aggregate measures of trust and political ideology over time. Of course, that’s a different study, but it feels that would answer the same questions much more convincingly.

My final criticism is that the discussion is too focussed on the U.S. Of course, issues of the interaction of population density, behaviour and disease spread are of relevance across the globe. The paper should attempt to speak to this audience rather than to focus on the U.S. rather narrowly.

Reviewer #2: This paper examines the relationship between individuals’ attitudes concerning COVID-19 and disease burden, and whether these relationships was significantly different in rural populations. To that end, the authors develop their own survey covering three states: Idaho, Texas, and Vermont. I think the authors have written an interesting paper on an important topic and while the literature is crowded, I do think that they make a contribution to it. However, I have some concerns and I feel like the current version comes up a bit short of robustness tests.

1) My first concern is about the measure of disease burden. The authors consider the number of cases from the beginning of the pandemic at county level as a proxy of disease burden. However, the prior epidemiology literature has used the number of deaths to better account for the spread of the COVID-19. Thus, I would suggest a robustness test where you make use of COVID-19 deaths.

1) The adoption of social distancing measures (including business closures and stay-at-home orders) may be influencing the trust-risk-behavior itself. As the authors highlight, the implementation of NPIs varied widely between communities within the US as they took place at distinct geographic levels (some at the county, others at the state) and for different periods of time. Thus, without appropriate controls it is hard to disentangle the effect of disease burden from the implementation of the NPIs. Can the authors control for the timing and intensity of the NPIs at county level?

3) I am not sure whether there is available data on the number of cases/deaths at city level. In this case, does it make sense to use disease data corresponding to the county instead of using city data when we know people addresses? If there is not available information I would suggest the authors to note that in the text.

4) I would suggest as another robustness test to amplify the set of demographic controls if it is possible. The authors describe the sample in terms of education and total household income; however, it is not clear to me whether these controls have been included in the model.

5) Are the estimates weighted?

6) I would suggest the authors to note the limitation of the low number of observations in the text.

7) Another important data limitation is that the survey period does not enable you to explore the existence of pre-trends during the months prior. Can the authors manage to assess that?

Reviewer #3: This is a nice paper, but somewhat too long compared with the contents. I am also puzzled by Model 2 (Supplemental Appendix page 8) that assumes that economic and health risk perceptions are affecting trust. I don't see what mechanism would make this plausible, or how it could be tested with the current data. If using data collected in one survey, I am worried that even if running formally regressions in line with Model 2, we woulf effectively capture correlational patterns the causal effect of which would go the other way. I suggest either dropping Model 2, or arguing more convincingly why it deserves to be kept and can be tested.

I think that streamlining the paper would improve its impact as potential readers would be more likely to read it through.

6. PLOS authors have the option to publish the peer review history of their article (what does this mean?). If published, this will include your full peer review and any attached files.

Reviewer #1: No

Reviewer #2: No

Reviewer #3: No

---

## [Author Response · Author response to Decision Letter 0]

8 Feb 2022

***NOTE THAT THE TEXT BELOW IS GIVEN IN OUR COVER LETTER IN AN EASIER TO READ FORMAT***

Dear editor(s),

We would like to thank the reviewers and editors for their helpful comments on our manuscript. In italic below, we give our responses to the comments provided by the reviewers after our responses to the editorial requests. In particular, we have attempted to clarify/respond to comments regarding our methods. In most of the cases where comments focused on variations of our model, we ran those analyses—as part of the exploratory phase—prior to our initial submission. We specifically now mention these prior analyses but are not presenting them in the revision. We do not present them in the revision for two reasons: First, they do not add to the results we already present, but they do act as a very weak form of cross-validation. Second, addition of numerous variations of the model (beyond the six we already investigate) will significantly add to the length of the manuscript and supplementary materials (which was already criticized by one reviewer); we also believe it would reduce the readability of our research by presenting various “dead ends” to readers. However, if the editors feel like the addition of more model variations is needed, we are happy to do so. Thank you again for the consideration of our work for publication in PLOS ONE.

Best,

Ben Ridenhour

Editorial Requests:

1. We believe all of our materials match the PLOS ONE styles that are specified.

2. We are unclear as to the discrepancy between the acknowledgments sections funding statement and the funding statement in our submission. It looks like we’ve referenced the NIH grant in both places (and the identical grant number), and it looks like we mention the Renfrew Fellowship (internal to University of Idaho) to Dr. Johnson-Leung that paid for undergraduate assistance on the work in both places as well. Can you give more explanation where the discrepancy is?

3. Our data should appear at https://doi.org/10.5061/dryad.0cfxpnw4c on Dryad once the paper is published. (We requested them be withheld until publication.)

4. Like (3), our minimal data set will appear on Dryad at the same DOI. 

Reviewer #1: Referee report for “Effects of Trust, Risk Perception, and Health Behaviour on Covid-19 Disease Burden: Evidence from a Multi-State US Survey

Summary:

This paper collects survey evidence from around 1000 individuals in Texas, Idaho and Vermont in November 2020, measuring demographic characteristics, social behaviours, risk perceptions, trust and political ideology. The paper assesses the relationship between these variables and case counts using SEM modelling. The paper finds that lower trust in rural locations offsets the natural advantages these places have for combating the spread of Covid-19.

Assessment

My main technical concern is with the implementation of the model. In many cases (e.g. Model 1A) the authors model behavioural intention as determining cases. The behavioural intention is then measured at the individual level, while case numbers are measured in the aggregate. But of course an individual’s behaviour is atomistic and has negligible effect on aggregate case numbers. So I don’t know how to interpret the estimates. I can’t recall ever seeing a model with an aggregate explained variable being driven by an individual-level explanatory variable. This approach should at least be discussed and supported.

RESPONSE: We are somewhat unclear about the reviewer’s comment on this. Virtually all statistical models take individual measurements and estimate a characteristic of the population (e.g., the mean, variance). Linear regression models typically take individual measurements and relate them to the mean of some other variable; a recent example of this from a similar COVID-19 epidemiological study would be Im and Kim (2021) [https://doi.org/10.3390/ijerph182312595]. 

From a higher-level perspective, we developed behavioral intention using the Health Belief Model (HBM) where an individual’s likelihood of adopting a given behavior is determined based on the perceived severity of disease and perceived effectiveness of recommended health behavior. In HBM behavioral intention is conceptualized as a psychological construct which is typically measured at the individual level and used to predict the group mean. 

More subjective concerns are as follows:

The main problem with this paper is the extremely small sample size. In each model the authors use 1000 observations to estimate 40 parameters. Obtaining significant and novel results in this way is only possible with rigid modelling.

RESPONSE: We were hoping to have a larger sample size given the number of surveys we originally mailed out (12,000). That being said, our sample size is still large enough to provide ample power to our statistical tests. For example, we found significant effect sizes (parameters) of ~0.05 in some of our models (see Tables S3.3 - S3.8). It is true that with greater sample size, we may have found even smaller effects to be significant (i.e., it is possible that we may missed some effects). In response to this comment and that of reviewer #2, we have added the following to the limitations paragraph in our discussion:

“Finally, while we received 1,087 responses out of the 12,000 surveys we sent out, having a larger sample size may have allowed us to attribute significant effects to other factors than those discussed here. That being said, the smallest significant effects in our models are of magnitude around 0.05, which suggests that our analyses are strong enough to detect small effects.”

My technical concern above leads me to another criticism, that much of questions here would be better addressed using aggregate data variation over both location and time. For example the google data can be leveraged to obtain detailed aggregate measures of behaviour across locations and over time. Similarly social surveys could be leveraged to obtain aggregate measures of trust and political ideology over time. Of course, that’s a different study, but it feels that would answer the same questions much more convincingly.

RESPONSE: We disagree that using aggregate data would be a better measure of population behavior. In prior COVID modeling efforts, the authors have used Google data to help determine contacts rates in various locales (particularly within Idaho). While these data are useful, they lack any detail on how might change their behavior as risk levels change. Rather, they reflect the effects of various public health orders (e.g., stay-at-home or mask mandates), socio-economic drivers, and other effects. Our method uses characteristics of individuals (e.g., political affiliation, gender, willingness to change behavior) to support a health belief model which has some predictive value on disease burden. These are data that one would not get from Google (e.g., whether self-declared liberal would be willing to stop shopping). It is perhaps an important distinction that we are looking to support/test health belief models and not just the best way to predict disease burden (with which one could use something like machine learning to do a better job). There is no way to understand risk perception, trust, and behavioral intention (and their relationships) using something akin to publicly available Google data. We agree with the reviewer that their suggestion would be a different study.

We now specifically add this statement in our introduction (line 47):

“We emphasize that our model is not an attempt to produce the best predictive model of COVID-19 burden, an effort which has been done using many other better suited methods for that task. Rather, we wish to determine a model of human behavior that could augment such models and increase their value to public health officials.”

We hope this clarifies the intention of our modeling efforts.

My final criticism is that the discussion is too focussed on the U.S. Of course, issues of the interaction of population density, behaviour and disease spread are of relevance across the globe. The paper should attempt to speak to this audience rather than to focus on the U.S. rather narrowly.

RESPONSE: We hesitate to speculate too much on regions outside of the US. Our survey respondents were only from 3 US states (Idaho, Texas, and Vermont), so even extrapolating to the entire US is perhaps speculative. However, we are more comfortable speaking about the US due to political, media, and cultural similarities within the US compared to say European countries where these factors may be totally different. We hope that, for example, our finding on politically conservative versus liberal individuals would hold for other regions/countries as well. If the editors feel like this speculation is warranted, we could add language to that effect.

Reviewer #2: This paper examines the relationship between individuals’ attitudes concerning COVID-19 and disease burden, and whether these relationships was significantly different in rural populations. To that end, the authors develop their own survey covering three states: Idaho, Texas, and Vermont. I think the authors have written an interesting paper on an important topic and while the literature is crowded, I do think that they make a contribution to it. However, I have some concerns and I feel like the current version comes up a bit short of robustness tests.

1) My first concern is about the measure of disease burden. The authors consider the number of cases from the beginning of the pandemic at county level as a proxy of disease burden. However, the prior epidemiology literature has used the number of deaths to better account for the spread of the COVID-19. Thus, I would suggest a robustness test where you make use of COVID-19 deaths.

RESPONSE: We agree that deaths due to COVID-19 are likely to be a better indicator of the severity of the pandemic impact in a given region. We further agree with the reviewer in that we did try a number of different disease measures for our models. For example, we tried using different cut-off dates (line 141 of the manuscript) to see if that had effects on our results, which they did not. We also tried using death counts instead of case counts, and we tried using the combination of the two (cases and death counts). In the latter case, the high degree of correlation between the two measures was prohibitive of their simultaneous inclusion into the model. Because many of the counties we worked with in this study are relatively small, there is a high degree of stochasticity and zero-observations when working with death counts which reduces statistical power. Thus, in the end, we opted to use case counts because of the reduced variance in the measure (and improved statistical inference). We did not include all of these exploratory analyses in our manuscript or appendix, but we would be happy to include them if so requested.

We have added a short paragraph starting on line 149 that explains our choice of using case counts over death counts.

1) The adoption of social distancing measures (including business closures and stay-at-home orders) may be influencing the trust-risk-behavior itself. As the authors highlight, the implementation of NPIs varied widely between communities within the US as they took place at distinct geographic levels (some at the county, others at the state) and for different periods of time. Thus, without appropriate controls it is hard to disentangle the effect of disease burden from the implementation of the NPIs. Can the authors control for the timing and intensity of the NPIs at county level?

RESPONSE: We agree with the reviewer’s points that a) recommendation by public health officials affects adoption and b) that adoption rates in turn affect disease burden. It is these very factors that we are trying to address with our models. In essence our model is using trust in public health, risk perception, and willingness to change behavior to predict adoption. We then use a measure of disease burden that is sufficiently far into the future (30 April 2021) to see the (predicted) effect on COVID-19 rates resulting from our health behavior model. That being said, if we actually knew (which we do not) when and where public interventions were enacted, we could probably increase the power of our model to predict disease burden. However, as mentioned before (see response to reviewer #1), the goal of the study was not to build the best predictive model of disease burden but to find a well-supported model for the behavioral side of human responses during the pandemic.

3) I am not sure whether there is available data on the number of cases/deaths at city level. In this case, does it make sense to use disease data corresponding to the county instead of using city data when we know people addresses? If there is not available information I would suggest the authors to note that in the text.

RESPONSE: There are not city level data available for the area we studied. We have noted this in the manuscript (line 137; this was also mentioned in line 423 of the original manuscript, now line 436).

4) I would suggest as another robustness test to amplify the set of demographic controls if it is possible. The authors describe the sample in terms of education and total household income; however, it is not clear to me whether these controls have been included in the model.

RESPONSE: We did originally include some of the other variables in our data set in the model (e.g., education and income). However, because they had little effect on the model as exogenous variables, they were not included in the “streamlined” models presented. As suggested by the reviewer, it is comforting to know that inclusion of such variables does not drastically alter the inferences made by our model. Again, we have not included these alternative models that were investigated along the way; if the editors feel like we should include these in the appendix in some way, we would be pleased to accommodate the request. 

To explain our choice, we have added the following statement on line 188:

“Other control variables---such as education and income---were originally explored in structural regressions as well; however, due to a lack of significant impact, these variables have been dropped from the presented analyses.”

5) Are the estimates weighted?

RESPONSE: They are not. (Though case burden was included as a rate per 100.) 

6) I would suggest the authors to note the limitation of the low number of observations in the text.

RESPONSE: See our response to reviewer #1 regarding sample size above. We have added text to the discussion regarding this point.

7) Another important data limitation is that the survey period does not enable you to explore the existence of pre-trends during the months prior. Can the authors manage to assess that?

RESPONSE: We are unsure what the reviewer means by “pre-trends.” (Pre-trends in what?) We do mention the limitation of the survey period our limitations paragraph of the discussion (line 445). If the trends in attitude are what the reviewer is suggesting, it seems unlikely that we would be assess such a trend. 

Reviewer #3: This is a nice paper, but somewhat too long compared with the contents. I am also puzzled by Model 2 (Supplemental Appendix page 8) that assumes that economic and health risk perceptions are affecting trust. I don't see what mechanism would make this plausible, or how it could be tested with the current data. If using data collected in one survey, I am worried that even if running formally regressions in line with Model 2, we would effectively capture correlational patterns the causal effect of which would go the other way. I suggest either dropping Model 2, or arguing more convincingly why it deserves to be kept and can be tested.

RESPONSE: We are happy the reviewer appreciates our work. Because we are using SEM to do our modeling, we specify the nature of the covariances within our models (i.e., which are zero and which must be fit) which results in the differences in pathways between the models. Thus, the worry about the causal effects “going the other way” is not warranted for the estimation procedure; that is not to say the arrows could not flow from trust to risk. In fact, that is exactly what models 1A and 1B are (Fig 1, Fig S3.1) and why we compare the fits of the models 1 to models 2 (and models 3). The premise behind model 2 is that, conditional on what individuals perceive as risks to their health or economic, they become more or less open to messaging from institutions such as CDC. For example, if one perceives their health or economic risk to be large, they may not care (trust) what any government agency tells them; conversely, if perceived risk is low, individuals may become much more open to messaging coming from the government. 

In order to provide more of an argument for the chosen models, we have reworded the paragraph that starts on line 192. We hope the wording now indicates that we are building on the work of others, such as the cited Siegrist (2019) paper which reviews the literature that explores the direction of causality between trust and risk perception. Hopefully the arguments presented in Siegrist (and the works therein) are convincing of the debate on the interplay between trust and risk perception.

I think that streamlining the paper would improve its impact as potential readers would be more likely to read it through.

RESPONSE: We are happy to cut portions of the paper if the editors feel it is too long and could be reduced in any particular/suggested way. Without particular suggestions as to what to remove, we are hesitant to do so (particularly given that other reviewers are asking for more material.)

---

## [Decision Letter · Decision Letter 1]

15 Mar 2022

PONE-D-21-36310R1Effects of trust, risk perception, and health behavior on COVID-19 disease burden: Evidence from a multi-state US surveyPLOS ONE

Dear Dr. Ridenhour,

Thank you for submitting your manuscript to PLOS ONE. After careful consideration, we feel that it has merit but does not fully meet PLOS ONE’s publication criteria as it currently stands. Therefore, we invite you to submit a revised version of the manuscript that addresses the points raised during the review process.

We look forward to receiving your revised manuscript.

Kind regards,

José Alberto Molina

Academic Editor

PLOS ONE

Journal Requirements:

Additional Editor Comments (if provided):

Dear Author/s,

I agree Rev 1.

Sincerely

Reviewers' comments:

Reviewer's Responses to Questions

**Comments to the Author**

1. If the authors have adequately addressed your comments raised in a previous round of review and you feel that this manuscript is now acceptable for publication, you may indicate that here to bypass the “Comments to the Author” section, enter your conflict of interest statement in the “Confidential to Editor” section, and submit your "Accept" recommendation.

Reviewer #1: (No Response)

Reviewer #2: (No Response)

2. Is the manuscript technically sound, and do the data support the conclusions?

Reviewer #1: Partly

Reviewer #2: (No Response)

3. Has the statistical analysis been performed appropriately and rigorously? 

Reviewer #1: Yes

Reviewer #2: (No Response)

4. Have the authors made all data underlying the findings in their manuscript fully available?

Reviewer #1: Yes

Reviewer #2: (No Response)

5. Is the manuscript presented in an intelligible fashion and written in standard English?

Reviewer #1: Yes

Reviewer #2: (No Response)

6. Review Comments to the Author

Reviewer #1: The responses to my points are mostly fine. With regard to my first point, the response of the authors is a little unfortunate; The Im and Kim paper that they refer to clearly uses aggregate variable as both dependent and independent variables. More precisely Im and Kim state "These 77 cities and counties were used as study units for the regression models." Accordingly I should be more precise in my question about the paper at hand: Is the unit of observation the location or the individual? If it's the location then you have very few data points. If it's the individual then I'm still not fully convinced by having an aggregate variable as the dependent variable.

On the other hand, the extended answer the authors give to my point is acceptable. But I would like to see a clear statement of the unit of observation.

Reviewer #2: (No Response)

7. PLOS authors have the option to publish the peer review history of their article (what does this mean?). If published, this will include your full peer review and any attached files.

Reviewer #1: No

Reviewer #2: No

---

## [Author Response · Author response to Decision Letter 1]

8 Apr 2022

Please see the attached cover letter for our response to the reviewer.

---

## [Decision Letter · Decision Letter 2]

27 Apr 2022

Effects of trust, risk perception, and health behavior on COVID-19 disease burden: Evidence from a multi-state US survey

PONE-D-21-36310R2

Dear Dr. Ridenhour,

We’re pleased to inform you that your manuscript has been judged scientifically suitable for publication and will be formally accepted for publication once it meets all outstanding technical requirements.

Kind regards,

José Alberto Molina

Academic Editor

PLOS ONE

Reviewers' comments:

Reviewer's Responses to Questions

**Comments to the Author**

1. If the authors have adequately addressed your comments raised in a previous round of review and you feel that this manuscript is now acceptable for publication, you may indicate that here to bypass the “Comments to the Author” section, enter your conflict of interest statement in the “Confidential to Editor” section, and submit your "Accept" recommendation.

Reviewer #1: All comments have been addressed

2. Is the manuscript technically sound, and do the data support the conclusions?

Reviewer #1: Yes

3. Has the statistical analysis been performed appropriately and rigorously? 

Reviewer #1: Yes

4. Have the authors made all data underlying the findings in their manuscript fully available?

Reviewer #1: Yes

5. Is the manuscript presented in an intelligible fashion and written in standard English?

Reviewer #1: Yes

6. Review Comments to the Author

Reviewer #1: (No Response)

7. PLOS authors have the option to publish the peer review history of their article (what does this mean?). If published, this will include your full peer review and any attached files.

Reviewer #1: No

---

## [Editor Report · Acceptance letter]

12 May 2022

PONE-D-21-36310R2 

Effects of trust, risk perception, and health behavior on COVID-19 disease burden: Evidence from a multi-state US survey 

Dear Dr. Ridenhour:

I'm pleased to inform you that your manuscript has been deemed suitable for publication in PLOS ONE. Congratulations! Your manuscript is now with our production department. 

Kind regards, 

on behalf of

Professor José Alberto Molina 

Academic Editor

PLOS ONE